# Hypertrophic Cardiomyopathy Complicated by Post-COVID-19 Myopericarditis in Patient with *ANO5*-Related Distal Myopathy

**DOI:** 10.3390/genes14071332

**Published:** 2023-06-24

**Authors:** Olga Blagova, Yulia Lutokhina, Marina Vukolova, Sergey Pirozhkov, Natalia Sarkisova, Dilara Ainetdinova, Anushree Das, Marina Krot, Vera Smolyannikova, Petr Litvitsky, Elena Zaklyazminskaya, Evgeniya Kogan

**Affiliations:** 1V.N. Vinogradov Faculty Therapeutic Clinic, I.M. Sechenov First Moscow State Medical University (Sechenov University), 119991 Moscow, Russia; blagovao@mail.ru (O.B.); sarkisovand@mail.ru (N.S.); dana_ya@mail.ru (D.A.); 2Department of Pathophysiology, I.M. Sechenov First Moscow State Medical University (Sechenov University), 119991 Moscow, Russia; mvoukolova@mail.ru (M.V.); arnheim-domain@yandex.ru (S.P.); litvitskiy_p_f@staff.sechenov.ru (P.L.); 3N.V. Sklifosovsky Institute of Clinical Medicine, I.M. Sechenov First Moscow State Medical University (Sechenov University), 119991 Moscow, Russia; anushree1das@gmail.com; 4Institute of Clinical Morphology and Digital Pathology, I.M. Sechenov First Moscow State Medical University (Sechenov University), 119991 Moscow, Russia; masolomahina@yandex.ru (M.K.); smva@bk.ru (V.S.); koganevg@gmail.com (E.K.); 5Laboratory of Medical Genetics, B.V. Petrovsky Russian Research Center of Surgery, 119991 Moscow, Russia; helenezak@gmail.com; 6N.P. Bochkov Research Centre for Medical Genetics, 119991 Moscow, Russia

**Keywords:** anoctamin 5, *ANO5*, NM_213599:c.2272C>T, hypertrophic cardiomyopathy, anticardiac antibodies, COVID-19, myopericarditis, myocarditis, arrhythmia, Miyoshi-like distal myopathy, heart failure

## Abstract

A 60-year-old male with hypertrophic cardiomyopathy, conduction disorders, post-COVID-19 myopericarditis and heart failure was admitted to the hospital’s cardiology department. Blood tests revealed an increase in CPK activity, troponin T elevation and high titers of anticardiac antibodies. Whole exome sequencing showed the presence of the pathogenic variant NM_213599:c.2272C>T of the *ANO5* gene. Results of the skeletal muscle biopsy excluded the diagnosis of systemic amyloidosis. Microscopy of the muscle fragment demonstrated sclerosis of the perimysium, moderate lymphoid infiltration, sclerosis of the microvessels, dystrophic changes and a lack of cross striations in the muscle fibers. Hypertrophy of the LV with a low contractile ability, atrial fibrillation, weakness of the distal skeletal muscles and increased plasma CPK activity and the results of the skeletal muscle biopsy suggested a diagnosis of a late form of distal myopathy (Miyoshi-like distal myopathy, MMD3). Post-COVID-19 myopericarditis, for which genetically modified myocardium could serve as a favorable background, caused heart failure decompensation.

## 1. Introduction

During recent years, the professional community of cardiologists has been actively discussing the role of inflammation in the clinical presentation of patients with various types of cardiomyopathies. There are two main concepts that describe the mechanism involved in the development of the phenotype in which myocarditis is combined with cardiomyopathy. The first of these concepts states that myocarditis is the trigger that switches on the abnormal genetic program leading to the development of cardiomyopathy [1].

The second concept, on the contrary, assumes that genetically modified myocardium serves as a favorable background for the accession of viral infection or autoimmune aggression with a consequent onset of myocarditis [2,3]. So, there is no consensus on what is primary in the combination of myocarditis and cardiomyopathy, although it is generally accepted that myocarditis increases the area of fibrous tissue, resulting in the development and progression of cardiac insufficiency and/or arrhythmia. Reports about the combined occurrence of hypertrophic cardiomyopathy and other myocardial diseases are few and present special clinical interest [4,5]. 

Cardiomyopathy may be part of a systemic myopathy based on mutations in the Anoctamin family of genes. The mutant variants of one of them, *ANO5*, are associated with proximal or distal muscle weakness and various cardiac symptoms from a reduced LV ejection fraction to ventricular premature beats [6].

Homozygous mutations in *ANO5*, a gene encoding anoctamin 5, a putative calcium-activated chloride channel, have recently been reported in patients with adult-onset myopathies or an isolated high-CK level, and may appear to cause complications in the form of dilated cardiomyopathy [7].

It is described in the literature that *ANO5* contains 22 exons. There are over 120 *ANO5* mutations linked to human diseases. The most common *ANO5* mutations are c.191dupA (p.N64KfsX15) in exon 5, followed by c.1898 + 1G > A in intron 17, c.2272C > T (p.Arg758Cys) in exon 20 and c.692G > T (p.Gly231Val) in exon 8, respectively [8].

Here, we report a case of a new phenotype in a carrier of *ANO5* mutation NM_213599:c.2272C>T: the combination of myopathy with hypertrophic cardiomyopathy, complicated by post-COVID-19 myopericarditis. 

## 2. Materials and Methods

A 60-year-old male was hospitalized in the cardiology department in June 2022 with complaints of shortness of breath during moderate physical exertion, weakness in the distal muscles of the lower extremities, more pronounced at the beginning of walking and somewhat decreasing with regular movement, and increased sweating. The patient underwent clinical examination including the collection of venous blood samples for DNA diagnostics, blood tests (full blood count, biochemical panel and high-sensitivity troponin T), immunoelectrophoresis of blood and urine proteins with the determination of light chains of immunoglobulins via immunofixation, electrocardiography (ECG), echocardiography (ECHO) and skeletal muscle biopsy (hematoxylin and eosin staining, Congo red staining, polarizing microscope).

Whole exome sequencing (WES) for the proband’s DNA was performed using a SureSelect Human All Exon V7 (Agilent Technologies, Santa Clara, CA, USA) followed by next-generation sequencing on an Illumina platform (MGISEQ-2000 (DNBSEQ-G400), San Diego, CA, USA). Reads were aligned to the human genome build GRCh37/UCSC hg19 and the variant calling with an automatic module EMSEMPLE_VEP, with the following analysis of the sequence variants using a custom-developed bioinformatics pipeline. Confirmation of genetic findings in the proband was performed via capillary Sanger resequencing on an ABI 3730XL DNA Analyzer according to the manufacturer’s instructions (Thermo Fisher Scientific, Waltham, MA, USA). The direct capillary Sanger resequencing with alternative oligoprimers was performed for the coding exons and flanking 100 bp of intronic areas of the *ANO5* gene to avoid an occasional loss of the second rare variant upon *ANO5* testing. Pathogenicity assessment of all variants confirmed via Sanger re-sequencing was performed according to ACMG (2015) criteria (on behalf of the ACMG Laboratory Quality Assurance Committee) [9]. Voluntary informed consent was obtained from the patient. The study protocol was approved by the Sechenov University Local Ethics Committee (protocol № 10–22 from 19 May 2022), Moscow, Russia. This study was conducted in accordance with the guidelines of the Declaration of Helsinki.

## 3. Results

### 3.1. Case Presentation

The family history of the patient was not burdened with myopathy or cardiomyopathy. He considered himself to be largely healthy, and coped with domestic chores relatively well. In 2015, he experienced a short episode of lower-extremity edema, although dyspnea was not marked. Diuretic therapy caused the regression of edema. Aggravation of the condition occurred again in 2019 when edema was accompanied by tension in the chest, palpitations, dyspnea and orthopnea in supine position, and so he was hospitalized for assessment. ECG for the first time showed the presence of atrial fibrillation, while echocardiography revealed moderate hypertrophy of the left ventricle walls up to 13 mm and dilation of the left side of the heart (EDD 5.9 cm), with an ejection fraction (EF) of 47%. A roentgenogram of the chest demonstrated features of bilateral hydrothorax. A blood test showed a 1.5-fold elevation in ALT and AST, LDH of 677 U/L, creatinine of 127 µmol/L and CPK on repeated measurements of 1592-3313-1850 U/L, and NTproBNP had increased to 508 pg/mL (N < 125 pg/mL). The patient was administered with an ACE inhibitor, β-blockers, digoxin, furosemide and apixaban, with a positive effect. 

The patient’s condition remained satisfactory up to the end of Dec 2021, when two weeks after he had contracted the new coronavirus infection, he developed the acute worsening of dyspnea and appearance of orthopnea. Roentgenogram of the left pleural cavity showed the presence of about 900 mL of fluid. Upon echocardiography, no significant changes were found as compared to those in 2019. Upon ECG, atrial fibrillation with a rate of 147 per min was registered. The patient was subjected to a single pleural puncture with fluid evacuation. The dose of loop diuretics was doubled, while the other medical prescriptions remained the same. Upon MRI scan of the heart in April 2022, there was late gadolinium enhancement (LGE) and increased contrasting of the parietal and visceral pericardium and their separation up to 4 mm, with LV EF of 53% and RV EF of 42%. The patient was suspected to have myopericarditis on account of acute decompensation after viral infection, moderately increased levels of immunoglobulin light chains and increased ESR, CRP and fibrinogen. These were the reasons to start medication of Metypred, 8 mg/d, with a further increase to 16 mg/d. Edema subsided but low tolerance to physical activity and general weakness persisted.

Upon admission, he had moderate ankle edema, his breathing was vesicular, 24 per min, and no rales were heard. The tones of the heart were arrhythmic, muffled and 94/min, and blood pressure was 100/70 mm Hg. The liver was not enlarged. A blood test revealed an increase in plasma CPK activity up to 576 U/L and an elevation of high-sensitivity troponin T to 38.2 ng/L. A further increase in troponin T levels was not observed, which allowed us to exclude the acute injury to the myocardium. The repeated histochemical analysis of the serum and urine proteins via the method of quantitative free light chain determination failed to confirm any monoclonal secretion. With the aim of evaluating the immunologic activity of myocarditis, the titer of blood anticardiac antibodies (Ab) was assessed: Ab to antigens (Ags) of cardiomyocyte nuclei—1:80 (N—abs); Ab to endothelial Ags—1:80 (N ≤ 1:40) (N ≤ 1:40); Ab to cardiomyocyte Ags—1:80 (N ≤ 1:40); Ab to smooth muscle Ags—1:320 (N ≤ 1:40); Ab to cardiac conductive fibers Ags—1:160 (N ≤ 1:40).

ECG registered atrial fibrillation with a frequency of ventricular complexes of 84/min. The remarkable findings consisted of a low voltage of the QRS complexes in standard leads and insufficient R-wave progression in thoracic leads (Figure 1). According to the results of daily ECG monitoring while the patient took bisoprolol and digoxin, the average heart rate during daytime was 86/min, at night it was 64/min, RR max was 3180 msec and RR min was 280 msec; four ventricular premature beats were registered. Echocardiography revealed hypertrophy of the left ventricle up to 14 mm, dilation of the left atrium (84 mL), ejection fraction of the left ventricle (45%), TAPSE of 1.4 cm and a minimal amount of fluid in the pericardial cavity. 

To exclude the presence of systemic amyloidosis, a biopsy of the skeletal muscle was taken (Figure 2). Microscopy of the biopsy sample showed sclerosis of the perimysium, moderate lymphoid infiltration, sclerosis of the microvessels, dystrophic changes and a lack of cross striations in the muscle fibers. Amyloid was not found after Congo red staining and polarizing microscopy. The observed changes suggested primary myopathy of unclear origin.

The patient was consulted by a medical geneticist. The presence of a mixed phenotype (hypertrophy of the left ventricle with a loss of contractile ability, atrial fibrillation, weakness of the distal skeletal muscles combined with increased plasma CPK activity and the results of the skeletal muscle biopsy) suggested a late form of distal myopathy. It was recommended to perform whole exome sequencing to look for mutations in the gene panels responsible for primary cardiomyopathy and myopathy.

### 3.2. Differential Diagnostics and Treatment Strategy

Taking into account the absence of stable and long-lasting arterial hypertension, hypertrophy of the myocardium up to 14 mm with a decrease in its contractility, absence of left ventricle dilatation, low voltage of the QRS complexes in ECG and a stable and prominent increase in plasma CPK (from 700 to 3300 U/L), in combination with symptoms of peripheral myopathy, it is reasonable to make a diagnosis of long-lasting symptomless primary cardiomyopathy with a mixed phenotype within a frame of neuromuscular disease that serves as a basis for arrhythmia. Data in favor of systemic AL-amyloidosis were not obtained. The risk factors of coronary artery disease in this case were male sex and older age; however, symptoms of angina and scarring of the heart (also taking into account the MRI results) were not observed. 

In addition, the patient suffered from post-COVID-19 myopericarditis, for which the genetically transformed myocardium could serve as a favorable background. The evidence for this diagnosis was provided by acute decompensation of heart failure, associated in time with a new coronavirus infection, as well as by the results of the heart MRI (LGE, the presence of fluid in the pericardium, LGE of parietal and visceral pericardium), troponin T elevation and the increased titer of the anticardiac antibodies despite the administration of methylprednisolone. A decline in the LV ejection fraction to 45% may be explained by myocarditis or by pathophysiologic mechanisms that operate in neuromuscular disease. Methylprednisolone may aggravate the course of peripheral myopathy, and so the further elevation of its dose was considered to be inappropriate. If the clinical effect of methylprednisolone on myocarditis for this patient is not enough, future treatment may additionally include cytostatic administration. Attempts to restore the sinus rhythm were thought to be impractical, taking into account the long duration of the atrial fibrillation (>3 years) and the stable normosystoly without ventricular ectopy that was achieved via a combination of bisoprolol and digoxin. 

The patient received 40 mg/d furosemide, 25 mg/d spironolacton, 5 mg/d bisopralol, 16 mg/d methylprednisolon, 10 mg/d eliquis, 80 mg/d febuxostat, 0.25 mg/d digoxin, 50 mg/d sacubitril + valsartan, 10 mg/d dapagliflozin and 20 mg/d omeprazole. The correction of therapy improved the condition of the patient and decreased dyspnea with ordinary physical activity, although walking was still accompanied by excessive sweating (possibly as a side effect of the methylprednisolone administration).

The results of the whole exome sequencing revealed the presence of the pathogenic variant NM_213599:c.2272C>T (p.Arg758Cys) in the homozygous state in the sequence of exone 20 of the ANO5 gene.

## 4. Discussion

Recessive mutations in *ANO5* are clinically presented by three phenotypes: limb-girdle muscular dystrophy type 2L (LGMD2L), Miyoshi-like distal myopathy (MMD3) and increased serum creatine kinase with variable exercise intolerance [6]. Patients with LGMD2L typically have asymmetric atrophy and weakness in quadricep and bicep muscles, while those with MMD3 complain of distal weakness and reduced muscle strength, especially in the calf muscles. 

The most common pathogenic mutations of *ANO5* are c.191dupA (exon 5) and c.2272C_T (exon 20) [10], and no correlations were found between the type of mutations and their clinical manifestation (phenotype) [11].

It is natural to expect cardiac pathology in the carriers of *ANO5* mutations, since this gene is expressed in cardiomyocytes and encodes a putative intracellular Ca^2+^-activated chloride channel. However, cardiac involvement was only reported in 10–30% of patients with anoctaminopathies, varying from subclinical arrhythmia to symptomatic cardiomyopathy [12]. It should be noted that prominent symptoms of cardiac insufficiency with left ventricular hypertrophy have not been described in mutant *ANO5* carriers before, while dilative cardiomyopathy is much more frequent. The only case of hypertrophic cardiomyopathy in a patient with anoctaminopathy described before was mentioned among a Dutch cohort of 105 LGMD patients [13]. However, in this study, whilst meticulous DNA-testing of the *ANO5* gene was performed, sarcomeric genes which could be responsible for the hypertrophic phenotype were not investigated. Hypertrophic cardiomyopathy prevalence varies from 1:200 to 1:500 in the population [14], and it is quite probable that 1 of 105 patients will have classical sarcomeric cardiomyopathy. On the other hand, in this case, we described whole exome sequencing whilst performing an analysis of cardiomyopathy-associated genes, and no mutations typical for hypertrophic cardiomyopathy were identified. In our patient, we observed moderate hypertrophy of the left ventricle, dilation of the heart and decreased ejection fraction, which were accompanied by overt symptoms of heart failure with lower-extremity edema and dyspnea up to orthopnea. Dystrophic changes in the heart tissue were confirmed by the presence of atrial fibrillation and ventricular premature beats, with no signs of coronary insufficiency. A late form of distal myopathy (Miyoshi-like distal myopathy, MMD3) in this patient was evidenced by hypertrophy of the left ventricle, with a decreased contractile ability, fibrillation of atria and weakness of the distal skeletal muscles in combination with increased plasma CPK activity, as well as the results of the skeletal muscle biopsy.

Myopericarditis in this case is a complication of COVID-19 infection. A series of cases of chronic-biopsy-proven post-COVID myoendocarditis with SARS-Cov-2 persistence and a high level of anticardiac antibodies have been reported before. Most of these patients benefited from immunosuppressive treatment [15]. In our case, we not only observed a high level of autoantibodies, but also MRI signs of myopericarditis, together with troponin T elevation. Moreover, acute heart failure decompensation happened exactly after COVID-19 infection. 

Whether the onset of myopericarditis in this patient was promoted by *ANO5*-mediated cardiomyopathy is a debatable question. If dystrophic changes in the skeletal muscles mirror the similar deterioration in the myocardium, it is presumable that sclerosis of the microvessels and the interstitial space could seriously compromise microcirculation and cause capillary–trophic insufficiency with a subsequent loss of the tissue immune capacity. Exposure to SARS-CoV-2 could lead to the direct injury of cells that carry ACE2 receptors on their surface. 

Alternatively, additional injury to cardiomyocytes (manifested in this patient by elevated CPK activity and high troponin T levels) caused by inflammatory mediators in the course of the systemic inflammatory response could result in the exposure of masked antigens in the external membrane and the initiation of immune autoaggression against the heart muscle fibers. The latter is supported by the presence of high titers of antibodies against the cardiomyocyte Ags, cardiomyocyte nuclei and the cardiac conductive fibers in the patient’s blood. Also, molecular mimicry between the spike protein of SARS-CoV-2 and the cardiac antigens cannot be excluded [16].

COVID-19 infection is accompanied by a massive generation of TNF-α and other potent proinflammatory mediators [17]. TNF-α not only exerts a cytotoxic effect but also inhibits the pumping ability of the heart. TNF-α and other proinflammatory mediators have negative inotropic effects, initiate cardiomyocyte apoptosis and may activate matrix metalloproteinases to trigger a matrix-degrading program [18]. In the described patient, post-COVID-19 myopericarditis, for which genetically modified myocardium could serve as a favorable background, caused the acute decompensation of cardiac insufficiency, demonstrated by a decrease in the LV ejection fraction and systemic blood pressure, tachycardia and tachypnea. 

## 5. Conclusions

Mutations in the *ANO5* gene cannot only lead to myopathy but also to cardiac pathology with a phenotype of hypertrophic cardiomyopathy. In cases of unexplained decompensation of heart function in patients with primary genetically determined cardiomyopathy, it is recommended to carefully identify and treat the accompanying myopericarditis. 

## Figures and Tables

**Figure 1 genes-14-01332-f001:**
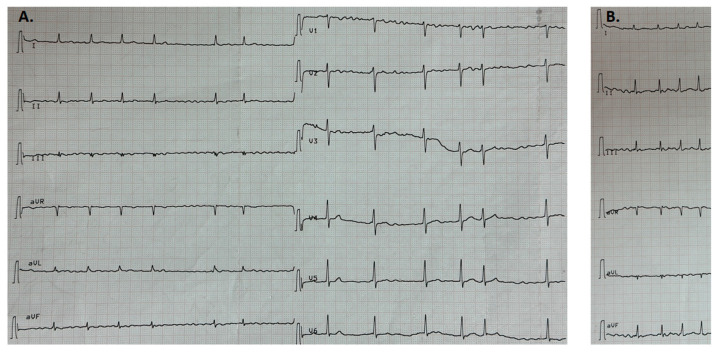
Electrocardiogram of the patient (descriptions in the text). (**A**) ECG at rest. (**B**) During the inspiration.

**Figure 2 genes-14-01332-f002:**
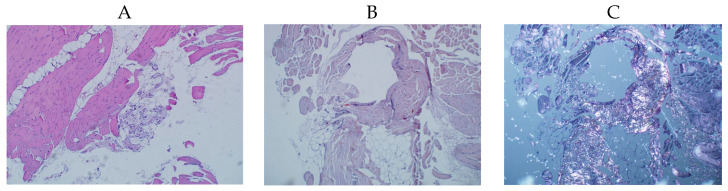
Results of the skeletal muscle biopsy (descriptions in the text). (**A**) Hematoxylin and eosin staining. (**B**) Congo red staining. (**C**) Polarized light microscopy.

## Data Availability

The data presented in this study are available upon request from the corresponding author.

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
