# Peer review of "Hypertrophic Cardiomyopathy Complicated by Post-COVID-19 Myopericarditis in Patient with ANO5-Related Distal Myopathy"

_genes, 2023, doi:10.3390/genes14071332_

Round 1
Reviewer 1 Report
The authors reported one case that a male old man with ANO5-related distal myopathy is complicated with hypertrophic cardiomyopathy. Since ANO5 gene is expressed in cardiomyocytes, it is surprising that a small fraction of patients with anoctaminopathy reported cardiac symptoms. The authors described the details about a patient’s presentation, diagnosis, and treatment. It was rare cases that a patient with anoctaminopathy had hypertrophic cardiomyopathy, especially developed myopericarditis after affected by COVID19. However, they should add one reference which also reported one case cases that a patient with anoctaminopathy had hypertrophic cardiomyopathy (Neuromuscul Disord. 2013;23:456–60). The authors should modify the according Discussion part.
Minor editing of English language required.
In line 102, is it 1.5-fold elevation of ALT?
Author Response
Point 1: The authors reported one case that a male old man with ANO5-related distal myopathy is complicated with hypertrophic cardiomyopathy. Since ANO5 gene is expressed in cardiomyocytes, it is surprising that a small fraction of patients with anoctaminopathy reported cardiac symptoms. The authors described the details about a patient’s presentation, diagnosis, and treatment. It was rare cases that a patient with anoctaminopathy had hypertrophic cardiomyopathy, especially developed myopericarditis after affected by COVID19. However, they should add one reference which also reported one case cases that a patient with anoctaminopathy had hypertrophic cardiomyopathy (Neuromuscul Disord. 2013;23:456–60). The authors should modify the according Discussion part.
Response 1:
Dear Reviewer,
Thank you for your discussion and precious comments!
During the literature review, we missed a case of hypertrophic cardiomyopathy in patient with anoctaminopathy you kindly pointed out and added it to the discussion of the article. However, in this study meticulous DNA-testing of ANO5 gene was performed, but sarcomeric genes which could be responsible for hypertrophic phenotype have not been investigated. Hypertrophic cardiomyopathy prevalence varies from 1:200 to 1:500 in dependence from population [Maron BJ et al. Diagnosis and Evaluation of Hypertrophic Cardiomyopathy: JACC State-of-the-Art Review. J Am Coll Cardiol. 2022 Feb 1;79(4):372-389. doi: 10.1016/j.jacc.2021.12.002] and it is quite probable that one of 105 patients has classical sarcomeric cardiomyopathy. On the opposite, in case we describe the whole exome sequencing with analysis of cardiomyopathy-associated genes was performed and no mutations typical for hypertrophic cardiomyopathy were identified.
Point 2: Minor editing of English language required.
In line 102, is it 1.5-fold elevation of ALT
Response 2: Thank you for the comment. It is exactly 1.5-fold elevation. We fixed it.

Reviewer 2 Report
In this manuscript, the authors reported a patient with hypertrophic cardiomyopathy with post-COVID myopericarditis. After whole-exome sequencing, one pathogenic variant NM_213599:c.2272C>T of the ANO5 gene has been identified, which may be caused the hypertrophic cardiomyophathy. It remained unclear that myopericarditis occurred before or after COVID. More biomarkers needs to be identified, such as immuno assays, including autoantibodies assay, cytokine assay. It would be great to test after post-COVID, whether autoantibodies has been generated which attack heart tissue or not.
Author Response
Response to Reviewer 2 Comments
Point 1: In this manuscript, the authors reported a patient with hypertrophic cardiomyopathy with post-COVID myopericarditis. After whole-exome sequencing, one pathogenic variant NM_213599:c.2272C>T of the ANO5 gene has been identified, which may be caused the hypertrophic cardiomyophathy. It remained unclear that myopericarditis occurred before or after COVID. More biomarkers needs to be identified, such as immuno assays, including autoantibodies assay, cytokine assay. It would be great to test after post-COVID, whether autoantibodies has been generated which attack heart tissue or not.
Response 1:
Dear Reviewer,
Thank you for your work and critical comments!
We added more information about the post-COVID myopericarditis in Differential diagnosis and Discussion sections. Diagnosis of post-COVID myopericarditis in the describe clinical case was based on acute decompensation of the heart failure associated in time with a new coronavirus infection, as well as by the results of the heart MRI (LGE, the presence of fluid in the pericardium, LGE of parietal and visceral pericardium), troponin T elevation and the increased titre of the anticardiac antibodies despite the administration of methylprednisolone. The last reflected high autoimmune activity. The patients was followed up by cardiologists before COVID-19. There were no signs of pericardial involvment of myocarditis during his previous hospitalizations, his heart failure decompensations were caused by hypertrophic cardiomyopathy and myocardial lesions in frames of anoctaminopathy. We would be glad to performe immuno assays and cytokine assay but we don’t have this technical opportunities. We assesed just anticardiac autoantibodies which is available in our clinic. In one of the studies of our research group we reported high levels of anticardiac antibodies in patients with biopsy-proven post-COVID myocarditis and that is we find it possible to use this test in complex diagnosis of post-COVID myocarditis without additional immunological studies [Blagova, O.; Lutokhina, Y.; Kogan, E.; Kukleva, A.; Ainetdinova, D.; Novosadov, V.; Rud`, R.; Savina, P.; Zaitsev, A.; Fomin, V. Chronic biopsy proven post‐COVID myoendocarditis with SARS‐Cov‐2 persistence and high level of antiheart antibodies. Clin Cardiol. 2022; 45, 952‐959].
